# Molecular basis of IRGB10 oligomerization and membrane association for pathogen membrane disruption

Hyun Ji Ha[1], Hye Lin Chun[1,2], So Yeon Lee[1,2], Jae-Hee Jeong[3], Yeon-Gil Kim[3] & Hyun Ho Park [1,2✉]

Immunity-related GTPase B10 (IRGB10) belongs to the interferon (IFN)-inducible GTPases, a family of proteins critical to host defense. It is induced by IFNs after pathogen infection, and plays a role in liberating pathogenic ligands for the activation of the inflammasome by directly disrupting the pathogen membrane. Although IRGB10 has been intensively studied owing to its functional importance in the cell-autonomous immune response, the molecular mechanism of IRGB10-mediated microbial membrane disruption is still unclear. In this study, we report the structure of mouse IRGB10. Our structural study showed that IRGB10 bound to GDP forms an inactive head-to-head dimer. Further structural analysis and comparisons indicated that IRGB10 might change its conformation to activate its membrane-binding and disruptive functions. Based on this observation, we propose a model of the working mechanism of IRGB10 during pathogen membrane disruption.

[1] College of Pharmacy, Chung-Ang University, Seoul 06974, Republic of Korea. [2] Department of Global Innovative Drugs, Graduate School of Chung-Ang University, Seoul 06974, Republic of Korea. [3] Pohang Accelerator Laboratory, Pohang University of Science and Technology, Pohang 790-784, Republic of Korea. ✉email: xrayleox@cau.ac.kr

Host defense against pathogen infection is critical for the survival of any organism; the evolutionarily conserved processes through which this defense is accomplished have been extensively studied owing to their important role[1–3]. The failure of human host defense systems leads to the spread of a variety of infectious diseases[4].

Interferon (IFN)-inducible GTPases are a family of host defense-related molecules that are specifically involved in resistance to bacterial infections[5]. Mx proteins[6], guanylate-binding proteins (GBPs)[7], very large inducible GTPases (VLIG)[8], and immunity-related GTPases (IRGs)[9] all belong to this family. The proteins of this family are induced by the IFNs released by the host cell right after pathogen infection occurs; they are known to play a role in the removal of the pathogens using their GTPase activity[4,10].

IRGs (also called p47 GTPases), are IFN-inducible guanine nucleotide-binding proteins that exhibit GTPase activity; IRGs are among the most abundantly expressed proteins during the initial stages of bacterial infection[9,11]. Thus far, 23 IRG genes have been identified in the mouse genome; meanwhile, only three putative IRG genes—truncated versions of mouse IRGs—have been identified in humans[12]. This family, comprising the IRGM (1–3), IRGA (1–8), IRGB (1–10), IRGC, and IRGD subfamilies, contains proteins composed of around 450 amino acids, with a molecular weight of around 47–48 kDa[12]. Sequence identity between same subfamily is around 32% for IRGA subfamily, 48% for IRGB subfamily, and 35% for IRGM subfamily, and between different subfamily family is around 12–15%.

Although the exact functions of all the proteins in the IRG family are not yet fully understood, some initial biochemical and functional studies of IRGM3, IRGB6, and IRGA6 have shown that they are involved in the cell-autonomous immune response against several pathogens, including *Toxoplasma gondii* (*T. gondii*)[12–14]. The anti-pathogenic function of IRGB6 and IRGA6 against *T. gondii* was found to be mediated by the destruction of parasitophorous vacuoles (PV), which are special membrane structures in the host cell where the pathogen can reside[14,15]. To disrupt the PV, IRGB6 and IRGA6 accumulate on the PV membrane (PVM), forming a filament-like aggregation structure whose formation is dependent on the presence of GTP[15,16]. The precise mechanism of IRG-mediated membrane disruption is still unclear. Among the 22–23 IRG proteins in the mouse, the structure of IRGA6 is the only available structure[17,18]. The structural and biochemical studies showed that IRGA6 formed different forms of dimer in solution, and the presence of GTP was critical for the further oligomerization of IRGA6 in vitro, which is functionally important[17–19]. Although a couple of dimeric form of IRG6 was introduced, functionally important filament-like structure of IRG family was not elucidated so far. Due to this limited structural information of IRG family, it is still difficult to understand the working mechanism of IRG family. Structural studies on dynamin superfamily members, such as Atlastin-1 and bacterial dynamin-like protein (BDLP), indicate that large-scale rearrangements between the GTPase domain and helical domain are critical for membrane binding and assembly[20,21].

IRGB10, one of the proteins in the IRGB subfamily, is a key player in the cell-autonomous immune response, involved in the destruction of pathogen membranes or pathogen-protecting membrane structures such as PVs[22–24]. The expression of IRGB10 is induced by the IRF-1 transcription factor, which is in turn activated by IFN-mediated signaling during infection[24]. The expressed IRGB10, along with another family of IFN-inducible GTPases, the GBPs, is recruited to the pathogen membrane and destroy it after forming a massive filament-like aggregate[23]. The lysis of the pathogen releases microbial products, including microbial DNA and lipopolysaccharides (LPS); the detection of these products triggers the formation of various types of inflammasomes, activating inflammatory caspases and launching the immune response against the pathogen[22,23].

Although IFN-inducible GTPases have been studied intensively due to their functional importance in the cell-autonomous immune response, the molecular mechanisms of IRG-mediated and GBP-mediated membrane disruption are still not fully understood. To shed light on these processes in IRGs, we studied the structure of mouse IRGB10, which is considered to play a major role in microbial membrane disruption during the cell-autonomous immune response. Our results revealed that IRGB10 binds to GDP and forms a head-to-head dimer, which may be an inactive form. In the GDP-bound form, the switch I and switch II loops of the GTPase domain were extremely flexible, whereas the P-loop was stably fixed by the interaction with the beta-phosphate of GTP. Structural analysis and comparisons also revealed that IRGB10 might be able to change to an active conformation to achieve further oligomerization for the microbe membrane lysis.

## Results

**IRGB10 forms a variety of homo-oligomeric complexes in solution.** The main structural feature of IRG family proteins is the possession of one GTPase domain located between two helical domains. Although IRG proteins perform a critical immune system function through their involvement in bacterial lysis, the working mechanism of this process is still little-understood, with limited in-depth information available on the structure of this family's proteins. To explore the structure of multi-form IRGB10, the full-length mouse IRGB10 cDNA, coding for a protein with 406 amino acids, was synthesized and cloned into the pET21a expression vector (Fig. 1a).

To produce homogeneous protein samples, we conducted size-exclusion chromatography (SEC) twice, consecutively; we then conducted additional ion-exchange chromatography after affinity chromatography (Fig. 1b, c). These purification processes generated two homogeneous protein samples corresponding to the monomer and the dimer sizes, which were used for crystallization (Fig. 1d); only the dimer-size sample was crystallized successfully. To analyze the exact stoichiometry of IRGB10 in solution, we calculated its absolute molecular mass in both the monomeric and the dimeric peaks from the SEC, after subjecting the samples to multi-angle light scattering (MALS). Interestingly, the MALS results suggested that the molecular mass of the dimer-size sample was around 300 kDa (Fig. 1e), indicating that IRGB10 forms a hexamer or a heptamer in solution, although the dimer-size particle (calculated to around 80–100 kDa) co-existed with the 300 kDa particle (Fig. 1f). We believe that dimeric form of IRGB10 on SEC further oligomerized as time passed. According to MALS, the molecular size of the monomer-size sample was 50 kDa, indicating that IRGB10 in this peak was a monomer (Fig. 1g). In a former study of another IRG-family protein, IRGA6, it was shown that its dimerization and further oligomerization was a GTP-dependent reaction[25]. Because we did not add GTP during the purification steps, the oligomerization of IRGB10 observed in our results might be somewhat different from that of IRGA6 or endogenous GTP in the bacteria used for protein expression was used for the oligomerization of IRGB10. Overall, this experimental result showed that IRGB10 forms a variety of different homo-oligomeric complexes in solution.

Because the oligomerization of many proteins is known to be dependent on their concentration[26], we also tested the concentration-dependence of IRGB10 oligomerization using SEC. This experiment showed that IRGB10 formed more

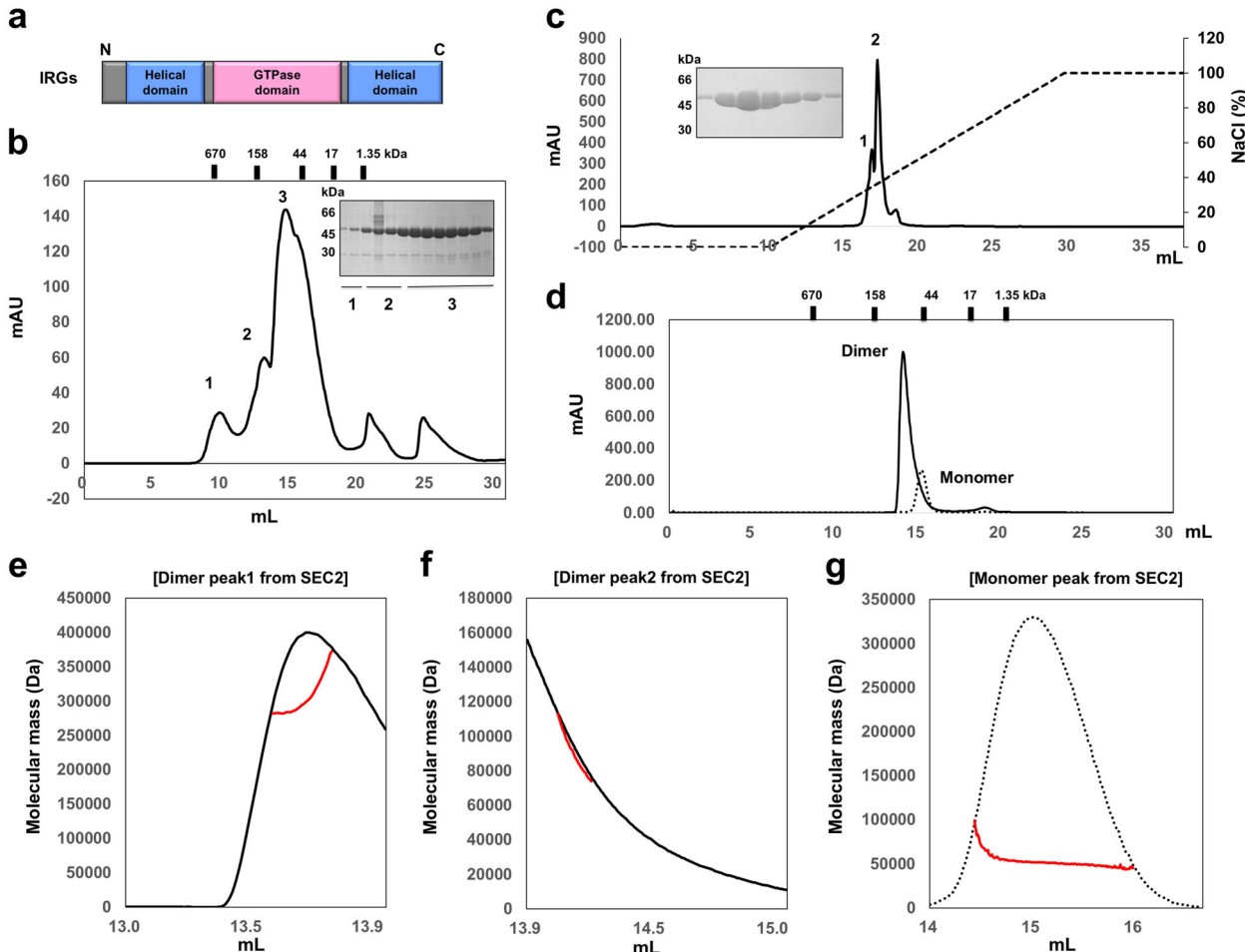

**Fig. 1 Purification and characterization of IRGB10. a** Domain boundary of the immunity-related GTPase (IRG) family; N and C indicate the N-terminus and the C-terminus, respectively. **b** Profile of the first size-exclusion chromatography (SEC). Three peaks were labeled in order. SDS–PAGE for the assessment of identity and purity was provided at the right side of the main peak. Loaded fractions are indicated by black bars with numbers. **c** Profile of the ion-exchange chromatography. Two peaks, 1 and 2, were labeled in order. Black dotted line indicates the gradient of NaCl. **d** Profile of the second SEC. **e**–**g** Multi-angle light scattering (MALS) profiles derived from the second peak of the first SEC (**e**), the third peak of the first SEC (**f**), and the monomer peak of the second SEC (**g**). Red line indicates the experimental molecular mass.

oligomeric complexes at higher concentration levels, indicating that IRGB10 oligomerization and filament formation is concentration-dependent (Supplementary Fig. 1).

**Overall structure of mouse IRGB10.** The limited knowledge of the structure of IRG-family proteins is often attributed to their tendency to form filament-like aggregates and insolubility[25]; however, we found that codon-optimized, full-length IRGB10 was soluble in solution, and a high amount of protein was purified and crystallized. The 2.6 Å crystal structure of the full-length mouse IRGB10 was solved and refined to $R_{work} = 20.22\%$ and $R_{free} = 27.67\%$. The crystallographic and refinement statistics are summarized in Table 1.

The structure exhibited the typical fold of the IRG family, containing two helical domains and one GTPase domain (Fig. 2a). The GTPase domain consisted of six β-sheets (S1–S6) and six α-helices (H4–H9). The helical domain consisted of 11 α-helices, H1–H3 from the N-terminus region and H10–H17 from the C-terminus region. The crystal structure showed that there were two molecules in the asymmetric unit (ASU), molecules A and B (Fig. 2b). The model of each molecule was constructed, from residue 15 to residue 406 for molecule A, and 14–406 for molecule B. The residues LEHHH at the C-terminus, which were from the

plasmid construct, were included in both models. Thirteen amino acids from the N-terminus were not visible in the model, and several loops, including switch I and II in GTPase domain, could not be constructed due to poor electron density. This absence of N-terminus structure was also observed in the structural study of IRGA6, indicating that the N-terminus loop containing 13 residues is a flexible and unstructured region[17]. The final model contained residues 15–101, 108–131, 134–160, 163–217, and 222–411 for molecule A and residues 14–101, 107–217, and 222–411 for molecule B. Although GTP and GDP were not added during the purification and crystallization steps, GDP was detected in the GTPase domain near the P-loop, indicating that endogenous GDP was incorporated after being expressed in bacteria (Fig. 2b, c). This is not surprising, as the IRG family affinity for GDP is known to be much higher than that for GTP[17,27]. The two molecules in the ASU formed a head-to-head dimer via GTPase domain (Fig. 2b). The initial molecule search for MR found two molecules formed head-to-tail dimer, which might be another candidate functional dimer (Fig. 2d, e). Although several models for dimerization in the IRG family have been proposed[17,18], the most recent structural study of IRGA6 suggested that a GTPase domain-mediated head-to-head dimer is a meaningful and functional dimer; it is similar to the head-to-head dimer formed by IRGB10 identified in our study (Fig. 2b)[18].

**Table 1 Data collection and refinement statistics.**

| Data collection | |
| --- | --- |
| Space group | P12(1)1 |
| Unit cell parameter a, b, c (Å) | |
| a, b, c (Å) | a = 62.51, b = 63.19, c = 117.92 |
| α, β, γ (°) | α = 90, β = 98.71, γ = 90 |
| Resolution range (Å)[a] | 29.14-2.6 (2.693-2.6) |
| Total reflections | 198,908 (20,437) |
| Unique reflections | 28,239 (2784) |
| Multiplicity | 7.0 (7.3) |
| Completeness (%)[a] | 99.82 (99.82) |
| Mean $I/\sigma(I)$[a] | 17.57 (3.95) |
| $R_{merge}$ (%)[a,b] | 7.63 (43.01) |
| Wilson B-factor (Å²) | 44.99 |
| *Refinement* | |
| Resolution range (Å) | 29.14-2.6 |
| Reflections | 28,215 (2784) |
| Reflections used for $R_{free}$ | 1756 (171) |
| $R_{work}$ (%) | 20.22 (25.81) |
| $R_{free}$ (%) | 27.67 (35.37) |
| No. of molecules in the asymmetric unit | 2 |
| No. of non-hydrogen atoms | 6343 |
| Macromolecules | 6244 |
| Ligands | 56 |
| Solvent | 71 |
| Average B-factor values (Å²) | 53.06 |
| Macromolecules | 53.1 |
| Ligands | 63.71 |
| Solvent | 42.64 |
| *Ramachandran plot* | |
| Favored/outliers (%) | 99.74/0.26 |
| Rotamer outliers (%) | 0 |
| Clashscore | 5.17 |
| RMSD bonds (Å)/angles (°) | 0.0083/0.963 |

[a]Values for the outermost resolution shell in parentheses.
[b]$R_{merge} = \Sigma_h \Sigma_i |I(h)_i - \langle I(h) \rangle| / \Sigma_h \Sigma_i I(h)_i$, where $I(h)$ is the observed intensity of reflection $h$, and $\langle I(h) \rangle$ is the average intensity obtained from multiple measurements.

**IRGB10 forms a head-to-head dimer mediated by the GTPase domain.** To identify the functional dimer of IRGB10, which is critical for achieving bacterial lysis, we analyzed the protein–protein interaction (PPI) in both the head-to-head dimer and the head-to-tail dimer using the PDBePISA PPI-calculating server[28]. According to this calculation, the interface of the head-to-head dimer was scored 0.25 in complex formation significance score (CSS), which ranges from 0 to 1 as interface relevance to complex formation increases, while the interface of the head-to-tail dimer was scored 0.00 in CSS. This result implies that the head-to-head dimer might be the form that is generally formed in solution, while the head-to-tail dimer might be formed by crystallographic packing. In the head-to-head dimer, a total dimer surface buries 1380 Å (a monomer surface area of 690 Å), which represents 3.5% of the total surface area, indicating that the PPI interface formed by the head-to-head dimer is wider than that formed by the head-to-tail dimer, although a 3.5% buried interface is not particularly large compared with a typical PPI. The main forces responsible for the formation of this dimeric interface, generated by the GTPase domains of both IRGB10 molecules, are massive hydrogen bonds and salt bridges (Fig. 3a). K193 from one molecule forms salt bridges with D185 and E233 from the other molecules; meanwhile, E94, D185, S186, N190, K193, S197, and E233 are involved in the formation of massive hydrogen bonds in the dimeric interface. In the case of the head-to-tail dimer, a total dimer surface buries 692 Å (a monomer surface area of 346 Å), which represents 1.8% of the total surface

area calculated by PDBePISA. The main interaction forces for this head-to-tail interaction are a salt bridge formed between K135 in one molecule and E206 in the opposite molecule, and hydrogen bonds formed by T132 and S96 from one molecule and N200, E202, and K405 from the opposite molecule (Fig. 3b).

Based on the dimeric interface analysis, we performed a mutagenesis study to identify which of the dimeric structures is more realistic in solution. Since D185 and K193 are the main interface residues in the formation of the head-to-head dimer, they were mutated to arginine and glutamic acid, respectively, producing D185R and K193E mutants. T132 and E206 residues, critical to the formation of the head-to-tail dimer, were also mutated to tryptophan and lysine, respectively, producing T132W and E206K mutants, which were expected to disrupt the head-to-tail dimer. We analyzed the effect of the mutations on the formation of the dimers using SEC. As indicated in Fig. 3c, d, although T132W and E206K did not affect dimer formation, D185R and K193E mutants had a definite disruptive effect, producing a new monomer peak in the SEC profile. Interestingly, loss of one of these salt bridges through mutation of D185R has a more significant impact on the ability of IRGB10 to dimerize than loss of both salt bridges through mutation of K193E (Fig. 3a, c). We did SEC experiments with those mutants several times and we got the same result showing that D185R has more strong effect on the disruption of IRGB10 dimer. Since K193E mutant formed more higher oligomeric peak (Fig. 3c), this phenomenon might be because of the solubility issue. Because K193E mutant becomes less soluble, disrupted dimer might go to higher oligomer (or aggregation) fraction. The molecular mass of the newly generated tentative dimer and monomer peaks from mutagenesis of D185R and K193E was further calculated by MALS to confirm the mutagenesis effect. MALS results showed that the molecular mass of the dimer-size sample produced by D185R was 92.5 kDa (Fig. 3e), whereas the molecular weight of the monomer-size sample was 49.8 kDa (Fig. 3f). MALS data for D193K mutant also produced similar result (Fig. 3g, h), indicating that newly produced monomer-size peak generated by head-to-head disruption mutants (D185R and K193E) is real IRGB10 monomer that is produced by disruption of head-to-head dimer interface. These results strongly suggest that IRGB10 forms a head-to-head dimer mediated by the GTPase domain.

**Comparison of the structure of IRGB10 with IRGA6.** The IRG family contains the highly conserved sequence, GXXXXGKS, in the G1 motif (switch I motif) of the GTPase domain (Supplementary Fig. 2). Several of the IRG family proteins, including IRGM1, IRGM2, and IRGM3, contain an atypical G1 motif sequence, GXXXXGMS. Based on this difference, the IRG family can be divided into two classes: GKS (GXXXXGKS) and GMS (GXXXXGMS)[12]. IRGB10 belongs to the GKS class and contains the sequence of GETGAGKS in its G1 motif (Supplementary Fig. 2).

In search of clues for inferring the mechanism of pathogen membrane disruption by IRGB10, we identified its structural homologs using the Dali server[29] and investigated their structure and function. The top three matches from a DALI search included IRGA6 (three different structures), BDLP, and Atlastin-1 (Table 2). IRGA6 is also a GKS class and the sequence identity between IRGB10 and IRGA6 was around 44%. Structural comparison by superposition showed that the position and length of several loops in IRGB10 differed from the equivalent loops in IRGA6, although the overall fold was the same, with a root mean square deviation (RMSD) of 2.5 Å over 406 Cα atoms (Fig. 4a). In particular, IRGB10 possessed a relatively long H15–H16 connecting loop and a well-defined H7–H8 connecting

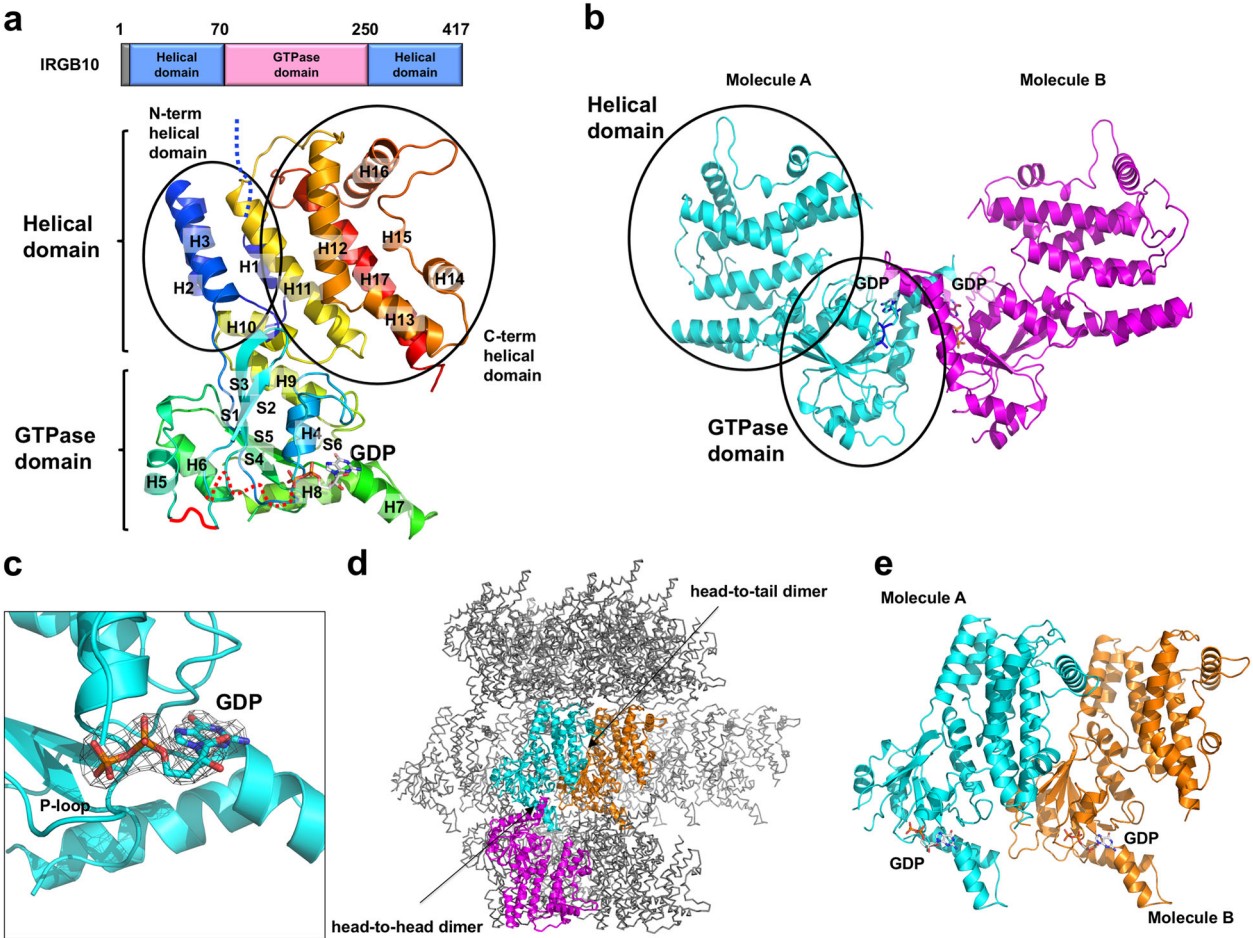

**Fig. 2 Crystal structure of the mouse IRGB10. a** The domain boundary and overall structure of IRGB10. The relative positions of the helical domain and the GTPase domains are shown in the bar diagram at the top. The multi-colored cartoon representation of monomeric IRGB10 is shown underneath. The chain from the N- to the C-terminus is colored blue to red. Helices and sheets are labeled with H and S, respectively. **b** A cartoon representation of two IRGB10 presented in an asymmetric unit. **c** Close-up view of the GDP bound in the nucleotide-binding pocket in the GTPase domain of IRGB10. $2F_o-F_c$ electron density map contoured at the $1\sigma$ level around GDP is indicated by the gray mesh. **d** Crystallographic packing analysis by searching for symmetry molecules. Dimeric molecules formed by head-to-head and head-to-tail interfaces are indicated by the color cartoons, while the other symmetry molecules are indicated by the gray ribbon structures. **e** Another tentative dimer structure with a head-to-tail interface, generated by the symmetry analysis.

loop, which were not visible in the structure of IRGA6 (Fig. 4a). Pair-wise structural comparison by superposition revealed that the helical domain of IRGB10 was tilted to 10.5° compared to the helical domain of IRGA6 (Fig. 4b). In the presence of GDP, the switch I and II loops of IRGB10 could not be constructed in the structural model due to poor electron density, while the P-loop was well-constructed and the density was clear. This indicates that the P-loop can be fixed by GDP, while the flexibility of switch I and II loops is independent of GDP incorporation in the nucleotide-binding pocket of IRGB10 (Fig. 4c).

It has been previously found that IRGA6 also forms dimers and further oligomeric complexes[16,25]. Although the structure of a functional higher oligomeric form has not been elucidated yet, a couple of dimeric strategies of IRGA6 have been suggested. An initial structural study of wildtype mouse IRGA6 suggested a symmetric dimer formed by the GTPase domain and the helical domain (Fig. 4d)[17]. However, a later study of mutant mouse IRGA6 indicated that it formed a GTPase domain-mediated symmetric dimer, similar to the head-to-head dimer of IRGB10 found in our study (Fig. 4d)[18].

The structure of head-to-head dimer of IRGB10 is similar with the head-to-head dimer of IRGA6 in that dimerization is mediated by GTPase domain. The side view of those two dimeric structures are nearly identical (Fig. 4e, f). However, the detailed dimerization strategy is different between two dimers. The main contact point of IRGB10 dimer is formed by H7 and connected loop of GTPase domain, whereas the dimeric interface of IRGA6 is mainly formed by P-loop and switch I of GTPase domain (Fig. 4e, f). Superposition of dimeric IRGB10 with that of IRGA6 more clearly showed that the dimeric structure of IRGB10 is different from the dimeric structure of IRGA6 (Fig. 4g). This different dimerization strategy might indicate the functional diversity of IRG family.

## Discussion

The head-to-head dimer formed by the GTPase domain of IRGB10, along with the similar, previously solved head-to-head dimer of IRGA6, suggest that this GTPase domain-mediated head-to-head dimer might be the main functional building block used by the IRG family for the membrane disruption of pathogens. However, a current dimeric structure is not sufficient for explaining how these head-to-head dimers further assemble around a pathogen membrane and disrupt it. To clarify this, we generated a possible membrane-bound model of the IRGB10 dimer based on the previously reported finding that myristoylation

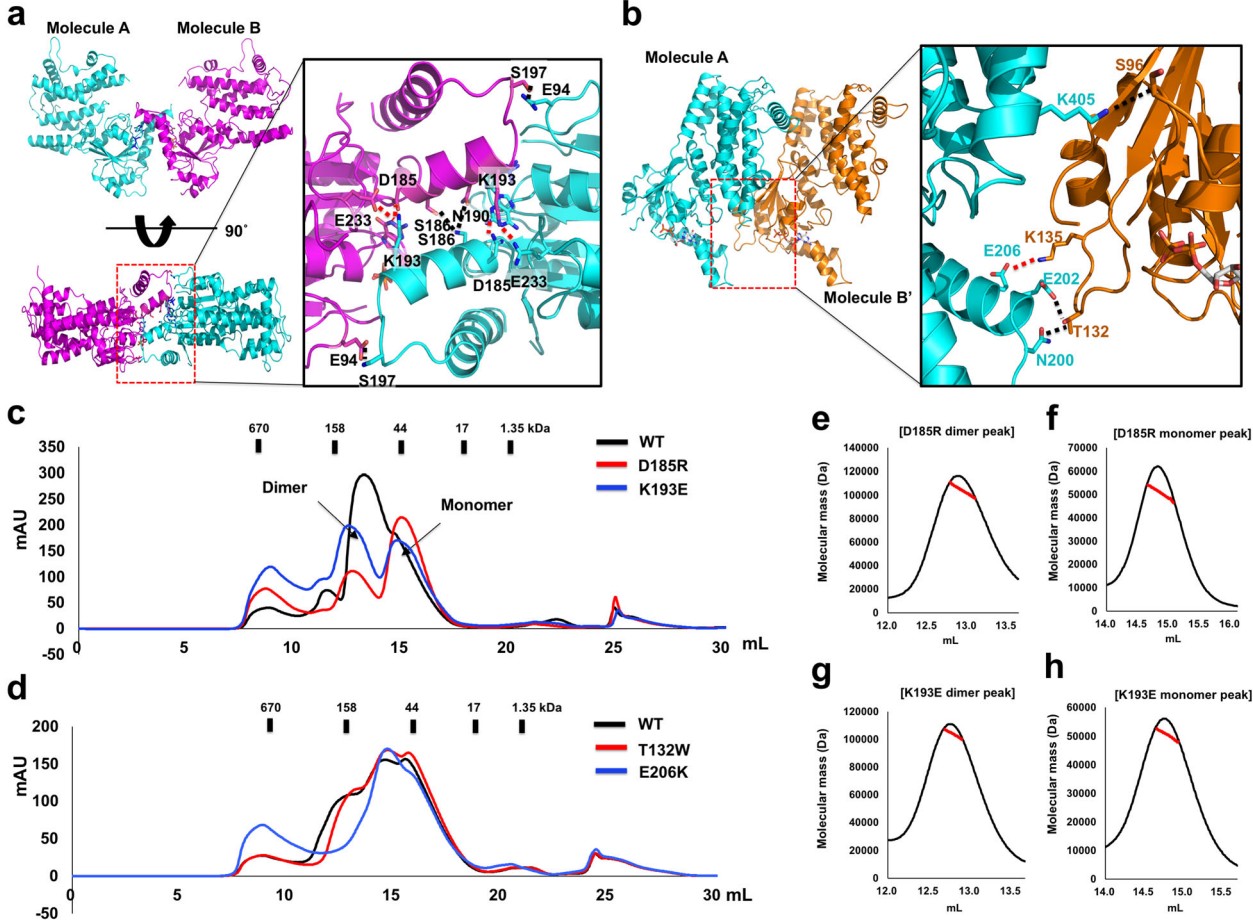

**Fig. 3 Dimeric structure of IRGB10. a** Details of the interface formed by the head-to-head dimer. Cartoon representation of the head-to-head dimer (left panel) with a close-up view of the main interface (right panel). Black dotted lines and red dotted lines indicate hydrogen bonds and salt bridges, respectively. **b** Details of the interface formed by the head-to-tail dimer. Cartoon representation of the head-to-tail dimer (left panel) with a close-up view of the main interface (right panel). Black dotted lines and red dotted lines indicate hydrogen bonds and salt bridges, respectively. **c** and **d** SEC profiles comparing the position of eluted peaks between wildtype IRGB10 and either (**c**) head-to-head dimer disruption mutants (D185R and K193E), or (**d**) head-to-tail dimer disruption mutants (T132W and E206K). **e** and **f** SEC-MALS analysis of newly produced peaks by D185R mutation. The tentative dimer peak (**e**) and monomer peak (**f**) were analyzed by MALS. **g** and **h** SEC-MALS analysis of newly produced peaks by K193E mutation. The tentative dimer peak (**g**) and monomer peak (**h**) were analyzed by MALS.

**Table 2 Structural similarity search using Dali[29].**

| Proteins (accession numbers) | Z-score | RMSD (Å) | Identity (%) | References |
|---|---|---|---|---|
| IIGP1 (1TPZ) | 44.2 | 2.5 | 44 | 17 |
| IRGa6 (5FPH) | 43.8 | 2.6 | 43 | 18 |
| IRGa6/ROP5B (4LV5) | 42.4 | 2.7 | 44 | 19 |
| BDLP[a] (2J69) | 14.4 | 3.9 | 21 | 30 |
| Atlastin-1 (3Q5E) | 14.3 | 3.5 | 20 | 20 |

[a]Bacterial dynamin-like protein.

modifications at the N-terminus (GQSSK and GAGKST sites, showed in Supplementary Fig. 2) and two tentative transmembrane regions (284EALKAGASATIPMMSFFND302 and 370AVTGGFVATGLYFRKSYY387) are important for the recruitment of IRGB10 to the pathogen membrane (Fig. 5a). This model showed that two membrane regions can be held together by the head-to-head dimer of IRGB10. In this case, the myristoylated N-terminus and both putative transmembrane regions were fixed in the pathogen membrane (Fig. 5b).

Even though this model can explain how IRGB10 fixes itself in the pathogen membrane, it does not provide an explanation for

how further oligomerization for membrane lysis is accomplished. Although a previous study has indicated that IRGB10 itself can disrupt the pathogen membrane and induce the further activation of the inflammasome[22], it has also been shown that IRGB10 works together with the GBP family to achieve bacteriolysis, especially with GBP5[23]. Therefore, we analyzed the direct interaction of mouse IRGB10 with mouse GBP5 using native PAGE. This experiment showed that IRGB10/GBP5 mixture failed to produce new complex band on the gel, which will be produced upon protein complex formation, indicating that IRGB10 might not get directly bound to GBP5 in either the absence or presence

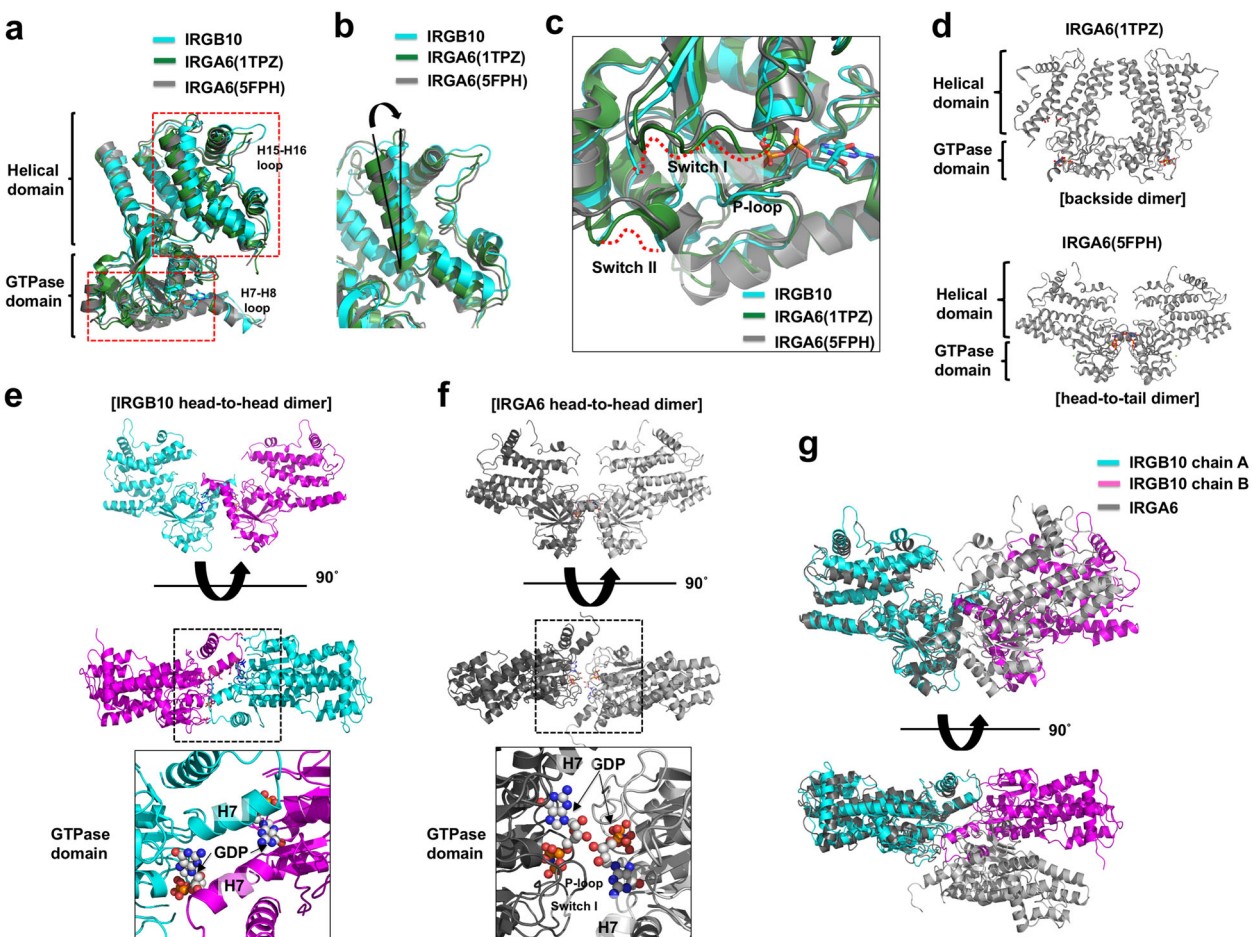

**Fig. 4 Structure comparison of IRGB10 with IRGA6. a** The superposition of IRGB10 with IRGA6. Two IRGA6 structures were used for the comparison. Red dotted line boxes indicate the two main regions that are magnified in (**b**) and (**c**). The position of two loops, the H7–H8 connecting loop (H7–H8 loop) and H15–H16 connecting loop (H15–H16 loop), whose structures are not identical to each other, are labeled. **b** Close-up view of the superposed helical domains. **c** Close-up view of the superposed GTPase domains. The unconstructed loops (switch I and switch II) are indicated by red dotted lines. The location of the P-loop is labeled. **d** Putative dimer structures of IRGA6 reported by previous structural studies. Two putative models are introduced. **e** and **f** Comparison of dimer interface of IRGB10 (**e**) with that of IRGA6 (**f**). Side view and top view of head-to-head dimer are shown at the upper and middle panels, respectively. Magnification of top view on GTPase domain are shown at lower panel. Black-dot box in the middle panels indicates the magnified site. **g** Superposition of IRGB10 head-to-head dimer with that of IRGA6. Cyan and magenta color indicate dimeric molecules of IRGB10, while the gray color indicates dimeric molecules of IRGA6.

of GTP/MgCl₂ in vitro (Fig. 5c; Supplementary Fig. 3). GBP2, another protein from the GBP family, also did not produce any complex band on the gel by mixing with IRGB10 (Supplementary Fig. 4). Although our protein interaction study indicated no direct interaction between IRGB10 and GBP family in vitro, based on the previous literature, it is clear that the ability of these proteins to act in concert depends on the infectious context[13,23]. Therefore, it cannot be ruled out that the interaction between IRGB10 and GBP family might be weak, or a specific cellular condition is needed for direct interaction. Since it has been shown that IRGB10 can solely disrupt the bacterial membrane without the binding and help of GBP-family proteins, the exact function of the GBP family in the process of bacteriolysis has yet to be uncovered. Sometimes cysteines and disulfide bonds might play a role in homo-oligomerization of proteins. To detect any surface exposed cysteines that might be involved in the formation of disulfide bonds for homo-oligomerization of IRGB10, we analyzed the sequence and structure of IRGB10. As the result, we found that there were four cysteines (C44, C212, C261, and C317) in the IRGB10 and all of them were not exposed to surface of IRGB10 structure (Supplementary Fig. 5), indicating that those

cysteines may not be involved in the formation of homo-oligomerization of IRGB10.

Owing to the structural similarity between IRGB10 and the GTPase domain-containing dynamin family, demonstrated by the Dali server analysis (Table 2), we analyzed the structure and oligomerization strategy of a member of this family and compared it with IRGB10. Like other proteins in the dynamin family, closely related to the IRG and GBP families, human atlastin 1 (Atl1) contains a GTPase domain and a helical domain (Fig. 5d). A previous structural study of Atl1 revealed that it also forms a dimer via the GTPase domain, although the dimerization interface differs from that of the IRGB10 dimer; it encompasses a much larger surface compared to IRGB10 (Fig. 5d)[20]. As described above, the IRGB10 head-to-head dimer via its GTPase domain was formed by hydrogen bonds and salt bridges formed between a limited number of residues; meanwhile, Atl1 forms a large hydrophobic and hydrogen-bonding network involving the switch I, switch II, G3, and G4 loops. One of the most interesting structural features of Atl1 is its ability to transform into an elongated form. During this process, its helical domain is rotated by 90° and extended to the opposite side of its GTPase domain

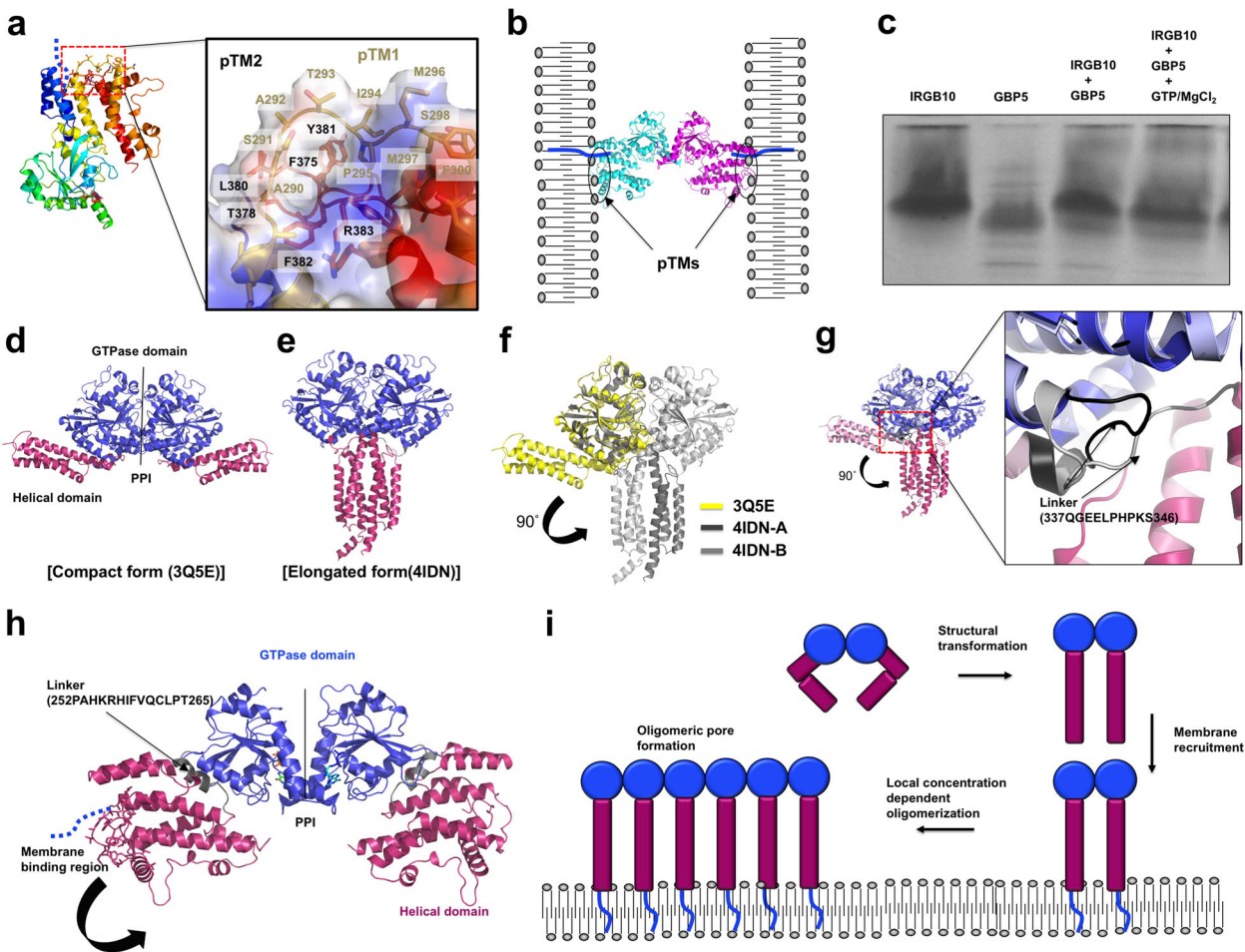

**Fig. 5 Putative model of IRGB10 oligomerization-mediated membrane disruption. a** The putative membrane-bound regions of IRGB10. The blue dotted line indicates the N-terminus region where myristoylation occurs. The close-up view in the panel on the right shows the two putative IRGB10 transmembrane regions (pTMs). The close-up view of this area is shown in a panel on the right. pTM1 and pTM2 indicate putative transmembrane regions 1 and 2, respectively. The electrostatic surface is represented for a better view of the hydrophobic pTMs. **b** A tentative model of membrane binding by the dimeric IRGB10 via a myristoylated N-terminus and two transmembrane regions (pTMs). **c** Native-PAGE. The loaded protein samples are indicated above each lane. **d** The compact-form structure of human atlastin 1 (Atl1). The two distinct domains are colored differently, with the GTPase domain in deep blue, and the helical domain in dark pink. PPI stands for protein–protein interface. **e** The elongated structure of Atl1. The color code used in (**d**) also applies here. **f** Superposition between the compact form (indicated in yellow) and elongated form (in gray) of Atl1. **g** The structure of dimeric IRGB10. The color code used for (**d**) and (**e**) also applies here. **h** Concentration-dependent oligomerization of IRGB10. The profile of SEC FPLC is provided. **i** Putative model of IRGB10 oligomerization via structural change and the resulting membrane disruption.

(Fig. 5e, f). This structural rearrangement of Atl1 is mediated by the linker region (residues 337–346) located between the GTPase and helical domains, composed of a loop and a small part of helix from the helical domain (Fig. 5g). During the structural transition, the small helix part of the linker region loses its helical structure and helps to elongate the helical domain (Fig. 5g). A similar linker region was also found in the structure of IRGB10 (residues 252–265), it was similarly located between the GTPase domain and the helical domain, and was also composed of a loop and a short helix (Fig. 5h). Considering the close connection between the dynamin family and the IRG family within the GTPase superfamily, it is possible that the helical domain of IRGB10 undergoes a similar structural change as that shown by Atl1. This structural change would extend the helical domain allowing the myristoylated N-terminus region and the putative transmembrane regions to be brought forward to the pathogen membrane for membrane disruption (Fig. 5h).

Based on the results of our structural analysis, we proposed a putative model of IRGB10-mediated pathogen membrane recruitment and oligomerization (Fig. 5i). In this model, we

hypothesize that the closed form of IRGB10, in complex with GDP, is its inactive state. Once GDP is exchanged for GTP, and followed by the hydrolysis of GTP, the IRGB10 helical domain is extended through elongation, which occurs using the mechanochemical force produced by the GTP hydrolysis. This structural change then reveals the membrane-binding region and the myristoylated N-terminus region for the recruitment of IRGB10 to the pathogen membrane. The increased local concentration of IRGB10 leads to further oligomerization. Although it is not yet clear precisely how the oligomerized IRGB10 disrupts the pathogen membrane, we consider it likely that GTP hydrolysis-mediated structural changes of the helical domain can result in a pore-forming conformation. In addition, two long loops, expected to serve as the transmembrane domain, might undergo structural changes in the presence of membrane lipids.

Unlike the dynamin family, no conformational changes of the helical domain were observed in the current IRGB10 structure. In the case of the BDLP, a member of another dynamin family in bacteria and similar to IRGB10, a closed conformation was observed in the crystal structures of the nucleotide-free and the

GDP-bound states[30]. Interestingly, another study observed a structural extension of the helical domain and wrapping of the membrane by further oligomerization of BDLP in the presence of lipid membrane using cryo-EM structure analysis[21]. If IRGB10 works in a manner similar to that of the dynamin family, there is a high possibility that structural change, GTP hydrolysis-mediated power generation, and further oligomerization-mediated membrane disruption happens only in the presence of phospholipid membrane.

## Methods

**Protein expression and purification.** The expression plasmid for full-length mouse IRGB10 corresponding to amino acids 1–417 was constructed by inserting the synthesized gene product, digested at the NdeI and XhoI restriction sites, into a pET21a vector. The gene sequence was derived from GenBank (ID: AGY29631) and gene synthesis was conducted using BIONICS (Seoul, Republic of Korea). The expression plasmid encoding the gene was delivered into *Escherichia coli* BL21 (DE3) cells using heat shock at 42 °C. A single recombinant colony was selected and cultured in lysogeny broth (LB) containing 50 μg/mL kanamycin overnight at 37 °C, after which the cells were transferred and cultured on 1 L large scale. When the optical density at 600 nm reached ~0.7, 0.5 mM isopropyl β-D-1-thiogalacto-pyranoside was added to the medium to induce gene expression, and the cells were further cultured for 18 h at 20 °C. The culture was harvested by centrifugation and the pellet was resuspended in 16 mL lysis buffer [20 mM Tris–HCl (pH 8.0), 500 mM NaCl, and 25 mM imidazole]. After adding a serine protease inhibitor, phe-nylmethanesulfonyl fluoride (Sigma-Aldrich, St. Louis, USA), the cells were disrupted by sonication on ice with eight bursts of 30 s each and a 90 s interval between each burst. The cell lysate was centrifuged at 10,000×*g* for 30 min at 4 °C to remove the cell debris. The supernatant was collected and mixed with nickel nitrilotriacetic acid (NTA) resin solution (Qiagen, Hilden, Germany) by gentle agitation overnight at 4 °C. The resulting mixture was applied to a gravity-flow column pre-equilibrated with lysis buffer. The column was washed with 100 mL of washing buffer [20 mM Tris–HCl (pH 8.0), 500 mM NaCl, and 60 mM imidazole] to remove unbound proteins. Then, 3 mL of elution buffer [20 mM Tris–HCl (pH 8.0), 500 mM NaCl, and 250 mM imidazole] was loaded onto the column to elute the bound protein. The resulting eluate was concentrated to 50 mg/mL and sequentially subjected to SEC. SEC purification was conducted using an ÄKTA explorer system (GE Healthcare, Chicago, USA) equipped with a Superdex 200 Increase 10/300 GL 24 mL column (GE Healthcare) pre-equilibrated with SEC buffer [20 mM Tris–HCl (pH 8.0), 150 mM NaCl]. The main protein fractions were pooled, concentrated to 30 mg/mL, and applied to a MonoQ column connected to the ÄKTA explorer system for ion-exchange chromatography. To generate the salt gradient, buffer A [20 mM Tris–HCl (pH 8.0)] and buffer B [20 mM Tris–HCl (pH 8.0) and 1 M NaCl] were gradually mixed with pump A and pump B in the ÄKTA explorer system. The peak fractions from ion-exchange chromatography were collected, concentrated, and re-applied to SEC. The peak fractions from SEC were pooled, concentrated to 12 mg/mL, flash-frozen in liquid $N_2$, and stored at −80 °C until use. The purity of the protein was assessed by SDS–PAGE.

**SEC-MALS analysis.** The absolute molar mass of mouse IRGB10 in solution was determined using MALS. The target protein, purified by affinity chromatography using NTA resin, was filtered with a 0.2 μm syringe-filter and loaded onto a Superdex 200 10/300 gel-filtration column (GE Healthcare) that had been pre-equilibrated in SEC buffer comprising 20 mM Tris–HCl (pH 8.0) and 150 mM NaCl. The mobile phase buffer flowed at a rate of 0.4 mL/min at room temperature. A DAWN-Treos MALS detector (Wyatt Technology, Santa Barbara, USA) was interconnected with the ÄKTA explorer system (GE Healthcare). The molecular mass of bovine serum albumin was used as a reference value. Data for the absolute molecular mass was assessed using the ASTRA program (Wyatt Technology).

**Crystallization and data collection.** For initial crystallization, 1 μL of protein solution in 20 mM Tris–HCl (pH 8.0) and 150 mM NaCl was mixed with an equal volume of reservoir solution, and the droplet was allowed to equilibrate against 500 μL of the mother liquor using the hanging drop vapor diffusion method at 20 °C. Crystals were initially obtained from a buffer comprising 20% (w/v) PEG8000 and 0.1 M HEPES (pH 7.5). The crystallization conditions were further optimized and finally adjusted to a buffer composition of 14% (w/v) PEG8000 and 0.1 M HEPES (pH 7.7). Diffraction-quality crystals appeared in 3 days and grew to a maximum size of $0.1 \times 0.1 \times 0.5$ mm³. For X-ray data collection, the crystals were soaked in the mother liquor supplemented with 40% (v/v) glycerol as a cryoprotectant solution, mounted, and flash-cooled in an $N_2$ stream at −178 °C. The diffraction data were collected at the Pohang Accelerator Laboratory with the 5C beamline (Pohang, Republic of Korea) at a wavelength of 10,000 Å. The diffraction data were indexed, integrated, and scaled using the HKL-2000 program[31].

**Structure determination and analysis.** The structure was determined by the molecular replacement (MR) phasing method using Phaser[32]. The previously solved mouse IRGA6 structure (PDB code 1TPZ), which has 44% amino acid sequence homology with IRGB10, was used as the search model. The initial model was built automatically with AutoBuild in Phenix and was further improved by manual building into $2F_o−F_c$ and $F_o−F_c$ electron density maps using Coot[33]. Model refinement was iteratively performed using phenix.refine in Phenix[34]: After a first rigid body refinement, 12 steps of refinement and rebuilding using PHENIX and COOT were performed. Each refinement step was composed of three mac-rocycles, and included, refinement of individual atomic coordinates, individual isotropic *B*-factors, and occupancies. When $R_{free}$ reached ~29%, water molecules were placed automatically by Phenix and subsequently examined manually for reasonable hydrogen bonding possibilities. Torsion/libration/screw motion restraints (TLS) parameters were applied at the final round of the refinement. The quality of the model was validated using MolProbity[35]. All the structural figures were generated using the PyMOL program[36].

**Mutagenesis.** Site-directed mutagenesis was conducted using a Quick-change kit (Stratagene) according to the manufacturer's protocols. Mutagenesis was then confirmed by sequencing. Mutant proteins were prepared using the method described above.

**Sequence alignment.** The amino acid sequences of the IRG family were analyzed using Clustal Omega (http://www.ebi.ac.uk/Tools/msa/clustalo/).

**Native page.** The formation of complexes between IRGB10 and GBPs was assayed by native (non-denaturing) PAGE conducted on a PhastSystem (GE Healthcare) with pre-cast 8–25% acrylamide gradient gels (GE Healthcare). Coomassie brilliant blue was used for the staining and detection of bands. IRGB10 was mixed with GBP5 or GBP2 in the absence and presence of GTP and $MgCl_2$, and incubated for 1 h at 4 °C, after which the mixture was subjected to electrophoresis. Complex formation was evaluated based on the appearance of newly formed bands or the disappearance of bands that were detected in single control protein bands.

**Reporting summary.** Further information on research design is available in the Nature Research Reporting Summary linked to this article.

## Data availability

Coordinates and structure factors have been deposited in the RCSB Protein Data Bank. The PDB ID code is 7C3K. Any other information is available from the corresponding author upon reasonable request.

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

## Acknowledgements

This study was supported by the National Research Foundation of Korea (NRF), which is funded by the Ministry of Education, Science, and Technology (NRF-2017M3A9D8062960, NRF-2018R1A2B2003635, and NRF-2018R1A4A1023822).

## Author contributions

H.H.P. designed and supervised the project. H.J.H. performed cloning, expression, and protein purification. H.J.H. and J.-H.J. crystallized and collected X-ray data. H.J.H., Y.-G.K., and H.H.P. solved the protein structure. H.L.C. and S.Y.L. performed MALS. H.H.P. and H.J.H. wrote the manuscript. All authors discussed the results, commented on, and approved the manuscript.

## Competing interests

The authors declare no competing interests.
