## [Peer Review File · Communications Biology]

Reviewers' comments:

Reviewer #1 (Remarks to the Author):

Ha et. al. presented a crystal structure of mouse IRGB10, a mouse interferon-inducible GTPase that mediates bacteriolysis in cell autonomous immunity. The authors found that like other IRG proteins, IRGB10 assumes multiple oligomerization states in solution. However, only the GDP-bound dimer form crystallized. GDP-bound IRGB10 assumed a head-to-head dimer organization, similar to GMPPNP-bound dimer form of IRGA6 reported previously. Using site-directed mutagenesis, Ha et. al. confirmed the head-to-head dimer is the stable dimer form in solution. By comparing IRGB10 structure with homologous GTPases structures, the authors propose that conformational change between GTPase domain and helical region, induced by GTP hydrolysis and membrane association, likely forms the basis of IRGB10-mediated bacteriolysis.

The manuscript presented a novel structure with strong accompanying biochemical results, the data are technically sound, and its results will be of interest to both immunologist and biochemists/structural biologists working in related areas.

Major issue:

The authors analyzed two possible dimer organization, the head-to-head dimer has both larger dimer interface and can be disrupted by mutagenesis in solution. Therefore, the head-to-head dimer is deemed the solution form. The head-to-tail dimer apparently is a crystal packing artifact. I suggest rearranging text and Figure 2 panels to reflect the fact the head-to-head dimer is the true dimer form and to be consistent with the section title.

Minor issues:

1. In Figure 4, a superposition of IRGB10 dimer and IRGA6 head-to-head dimer will assist in understanding the (dis)similarity between the two GTPases in dimeric form.
2. In Figure 1, MALS measurement of the dimer species is carried out on SEC1 peak 3, which potentially is contaminated by other homo-oligomers. A more precise measurement should be done on the purified dimer peak as shown in SEC2 (Figure 1D)
3. In Table 1, the difference between R_{work} and R_{free} is more than 7%, is there over-refinement?
4. Endogenous GDP was observed in the crystal structure, was magnesium (or metal ion) density also observed at the binding pocket?
5. The authors state that IRGB10 forms homo-oligomers in solution. Could cysteines and disulfide bonds play a role in that?
6. Page 4, lines 138-139, buffers A and B for MonoQ purification have the same composition.
7. Densities for switches I and II are missing in the IRGB10 structure (page 8). Are those regions on crystal packing interface?
8. Based on the crystal structure and surface characteristics, could the authors speculate on whether the two putative transmembrane regions (pTMs) penetrate membrane or only associate with one side of membrane?

Reviewer #2 (Remarks to the Author):

In this manuscript, the authors report the crystal structure of the mouse IFN-inducible GTPase IRGB10 to a resolution of 2.6 Å. Further, the authors explored the functional relevance of two differentially arranged IRGB10 dimers that were observed within the crystallographic structure and potential interactions between IRGB10 and guanylate-binding protein binding partners. The solving of the

crystal structure of IRGB10 and identification of potential homologues significantly enhances our understanding of this cell-autonomous protein. However, there are some issues with data interpretation and collection that need to be addressed or clarified before publication.

Major Points:

1) Fig 3C: The authors conclude that mutation of contact residues between the head-to-tail dimer does not disrupt the physiological dimer of IRGB10; this conclusion is based on size exclusion chromatography analysis of the molecular weights of different IRGB10 species in solution. However, based on the spectra, it appears that a higher concentration of the E206K mutant was loaded into the sample. Because the authors found that oligomerization of IRGB10 was concentration-dependent, the apparent disconnect between protein concentrations of the wild-type and mutant make it difficult to accurately conclude the effects of this mutation on the ability of the protein to oligomerize in solution.

2) Fig 3D and text: The authors state that K193 mediates the head-to-head dimerization of IRGB10 by interacting with both D185 and E233. However, loss of one of these salt bridges through mutation of D185 has a more significant impact on the ability of IRGB10 to dimerize than loss of both salt bridges through mutation of K193. Further, it appears that mutation of K193 had more of an impact on the formation of a higher molecular weight species. These observations need to be explained.

3) Based on native PAGE electrophoresis, the authors conclude that there is no interaction between IRGB10 and GBP2 or GBP5. However, based on the literature, it is clear that the ability of these proteins to act in concert depends on the infectious context. The evidence supporting the conclusion that these proteins do not directly interact to lyse bacteria therefore appears weak. The authors should consider modifying this conclusion in the text to make the caveats more apparent or utilize further experimentation needed to conclusively make this statement. Additionally, due to smears, multiple banding patterns, and a lack of indication of molecular weights, it is difficult to interpret this data.

Minor points:

1) Please update the text to clarify that IRGA6 is also a GKS GTPase. This will help strengthen the rationale for similarities between IRGB10 and IRGA6.

Reviewer #3 (Remarks to the Author):

IRGs comprise a family of dynamin-related, interferon-induced GTPases that have an important function in the destruction of intracellular pathogens. While 23 IRGs exist in mice, only a few truncated IRG members are present in human. IRGs are thought to oligomerize at the surface of pathogen containing vacuoles leading to their disruption, and the release and subsequent killing of the pathogen. Based on sequence comparisons, IRGs can be subdivided into IRGA, IRGB and IRGM members. So far, structural information has been described only for an IRGA member, but not for IRGB and IRGM members. In this manuscript, the authors report the crystal structure of the IRGB-family member IRGB10 in the GDP-bound state. Based on structure analysis and a comparison with other dynamin superfamily members, they propose a structural activation model for IRGs.

Besides some minor grammar mistakes, the manuscript is well written, figures are carefully prepared and the structural data appear sound. The structure itself and the dimerization interface is rather similar to that of the published Irga6 structures. Still, a structure of a representative IRGB family members may be useful for researchers in the innate immunity field.

I have a few concerns and a few suggestions to increase the scope of the work.

Introduction:

The introduction should contain more details on what is known from previous studies on the structure and biochemistry of IRG family members and the cellular function of IRGB10. Also a short paragraph on the structure of related dynamin-GTPases (in particular BDLP and atlastin) would be warranted to allow the reader a better assessment of the impact of this work. Furthermore, known differences in sequence and function of IRGA, IRGB and IRGM families should be worked out in detail.

Results:

Crystallographic data table

Rfree and Rwork differ by more than 7% - the structure may be overrefined. What was the refinement strategy (what exactly was refined)? Should a more conservative refinement be considered? Please include details of the refinement strategy in the Methods. Why were phenix and REFMAC5 used in combination?

Fig. 2: I would suggest first showing the structure of the IRGB10 monomer and then describe higher-order assemblies in the crystal. Please label the helical domains and GTPase domains in Fig. 2A. If I understand Fig. 2E correctly, chain A from one asymmetric unit forms a physiological dimer with chain B from the adjacent asymmetric unit? If this was the case, why not place and refine the physiological dimer in the asymmetric unit, rather than using the current placement obtained randomly by MR and likely representing a non-physiological dimer.

Fig. 2: The authors built a GDP molecule in the active site, although they did not add it during purification. Since the resolution of the structure is limited, they should prove by alternative methods that a GDP molecule is maintained during purification (by HPLC or mass spectrometry analysis). Can the GDP molecule be removed in the absence of magnesium, e.g. by a washing step with EDTA?

Fig. 3D: These data are not conclusive. Based on the UV signal, different concentrations were used for wt and the two mutants. Since IRGB10 shows a concentration-dependent assembly (Fig. S1), equal concentrations must be used for this experiment. Furthermore, MALS should be employed to analyze whether the second peak in the elution profile of the mutants indeed corresponds to a monomer. Do the mutants still contain bound GDP or do they lose it during purification in the absence of the G interface? How was the protein concentration determined? UV analysis may be misleading when nucleotides are bound differentially.

Fig. 3. The G interface described here is important for the story and should be further analyzed. For example, does nucleotide-free IRGB10 show a GTP-dependent assembly in solution that can be measured by turbidity (similar as for IRGA6)? In this case, wt and mutants should be tested in this assay. Does IRGA6 show a concentration-dependent GTPase activity that is disturbed for the mutants?

Fig. 4: The sequence alignment could be moved to the supplement. Instead, some more structural comparisons of the G domain dimer in IRGA6 and IRGB10 would be useful. For example, are residues involved in dimerization conserved in IRGB10 and IRGA6? Based on this, would one predict heterodimerization? A side-by-side comparison of the IRGB10 and IRGA6 dimers with details of the interaction would be useful.

Fig. 5: I found the introduced model of action interesting, but I would place it under the header 'Discussion', not 'Results'. The actual structure of a membrane-bound IRG may or may not look like the proposed model.

Line 280 - What does an interface score of 0.00 and 0.25 mean?

Line 324: I think it is sufficient to say that the closest relative of IRGB10 with a known structure is IRGA6, unless the authors want to describe the (presumably very small) differences in the three published IRGA6 structures.

Line 330 – root mean square deviation. Were all residues included in this analysis or only those that superimpose well ? Does the reported number correspond to Calpha superposition ?

Typos:

Abstract, line 39: it decorated by guanylate binding proteins (GBPs) directly targets ...
Please correct this sentence.

Line 57 IRG are an emerging family of host defense related ...
The first manuscript on IRGs dates back to 1998 (Boehm et al., J. Immunology), e.g. 'emerging' may be misleading.

Line 87: ... are recruited ...

Line 243: With great effort – Why with 'great effort'? Was there something particularly difficult in structure determination?

Dear Editor,

We have revised our manuscript based on the reviewers' suggestions. Please find our detailed responses to the reviewers' comments on the following pages.

Thank you very much for your time and effort in editing our manuscript, and we hope that it is now suitable for publication in "*Communications Biology*".

Sincerely,

Hyun Ho Park, Ph.D.
Professor of Pharmacy

Reviewers' Comments to Author:

Reviewer: 1

We thank the reviewer for his/her constructive comments on our work.

Comments (Major points):

1. The authors analyzed two possible dimer organization, the head-to-head dimer has both larger dimer interface and can be disrupted by mutagenesis in solution. Therefore, the head-to-head dimer is deemed the solution form. The head-to-tail dimer apparently is a crystal packing artifact. I suggest rearranging text and Figure 2 panels to reflect the fact the head-to-head dimer is the true dimer form and to be consistent with the section title.

Response: As reviewer's suggestion, we rearranged the text and Figure2 panels to reflect the fact the head-to-head dimer is the true dimer form. We also moved the figure 2B to figure 2A as suggested by reviewer #3. This rearrangement is way better to follow the structural details of IRGB10. We thank this reviewer for this suggestion.

2. In Figure 4, a superposition of IRGB10 dimer and IRGA6 head-to-head dimer will assist in understanding the (dis)similarity between the two GTPases in dimeric form.

Response: This is good comment. We totally agree with the reviewer's opinion. As suggested, we moved the sequence alignment to Supple info and added structural comparison of IRGB10 dimer with IRGA6 dimer.

3. In Figure 1, MALS measurement of the dimer species is carried out on SEC1 peak 3, which potentially is contaminated by other homo-oligomers. A more precise measurement should be done on the purified dimer peak as shown in SEC2 (Figure 1D)

Response: We agree with the reviewer's suggestion that stoichiometry of IRGB10 should be precisely measured. However, there is some misunderstanding on our measurement. We performed the MALS measurement of the dimer species using a dimer fraction collected on second SEC (Figure 1d). This purified dimer peak (Figure 1d) gave us the MALS results on figure 1e and 1f. We believe that dimeric form of IRGB10 on SEC further oligomerized as time passed. To avoid the reader's misunderstanding, we added this tentative oligomerization on the main text.

4. In Table 1, the difference between Rwork and Rfree is more than 7%, is there over-refinement?

Response: Although it is true that the structure was a little bit over-refined during the structural refinement process, we believe that the R values are acceptable range and the structure is not biased.

5. Endogenous GDP was observed in the crystal structure, was magnesium (or metal ion) density also observed at the binding pocket?

Response: When we refined our structure, we were not able to see the any ion density. Based on the reviewer's comment, we carefully analyzed the electron density around the GDP binding site, however, no clear ion density was observed.

6. The authors state that IRGB10 forms homo-oligomers in solution. Could cysteines and disulfide bonds play a role in that?

Response: We agree with this reviewer's opinion that cysteines and disulfide bonds might play a role in homo-oligomerization of IRGB10. With this comment, we analyzed the sequence and structure of IRGB10 to find any surface exposed cysteines that might be involved in the formation of homo-dimerization. As the result, we found that there were four cysteines (C44, C212, C261, and C317) in the IRGB10 and all of them were not exposed to surface of IRGB10 structure, indicating that those cysteines may not be involved in the formation of homo-oligomerization of IRGB10. We added this discussion at the sub-section of "**Tentative models of IRGB10 oligomerization and membrane association**" with new SI figure4.

7. Page 4, lines 138-139, buffers A and B for MonoQ purification have the same composition.

Response: Thank you for finding this incorrect information. We corrected it.

8. Densities for switches I and II are missing in the IRGB10 structure (page 8). Are those regions on crystal packing interface?

Response: No. They are not in crystal packing interface

9. Based on the crystal structure and surface characteristics, could the authors speculate on whether the two putative transmembrane regions (pTMs) penetrate membrane or only associate with one side of membrane?

Response: With current structure, it is hard to predict whether the two putative transmembrane regions (pTMs) penetrate membrane or only associate with one side of membrane. However, although previous biochemical study indicated that two tentative transmembrane regions (284EALKAGASATIPMMSFFND302 and 370AVTGGFVATGLYFRKSY387) might penetrate transmembrane, we think that IRGB10 only linked to membrane using myristoylation modifications and N-terminal region. The two putative transmembrane regions (pTMs) might only associate with one side of membrane based on our structural working model. We hope that we can determine this important issue by our next study.

Reviewer: 2

We thank the reviewer for his/her positive assessment of our work

Comments:

1. Fig 3C: The authors conclude that mutation of contact residues between the head-to-tail dimer does not disrupt the physiological dimer of IRGB10; this conclusion is based on size exclusion chromatography analysis of the molecular weights of different IRGB10 species in solution. However, based on the spectra, it appears that a higher concentration of the E206K mutant was loaded into the sample. Because the authors found that oligomerization of IRGB10 was concentration-dependent, the apparent disconnect between protein concentrations of the wild-type and mutant make it difficult to accurately conclude the effects of this mutation on the ability of the protein to oligomerize in solution.

Response: We totally understand the reviewer's concern. We re-performed all the SEC-experiments by using the same amount of loaded proteins. In addition, we also performed SEC-MALS to confirm the mutagenesis effect. Please see the new figure 3.

2. Fig 3D and text: The authors state that K193 mediates the head-to-head dimerization of IRGB10 by interacting with both D185 and E233. However, loss of one of these salt bridges through mutation of D185 has a more significant impact on the ability of IRGB10 to dimerize than loss of both salt bridges through mutation of K193. Further, it appears that mutation of K193 had more of an impact on the formation of a higher molecular weight species. These observations need to be explained.

Response: We did SEC experiments with those mutants several times and we got the same result showing that D185R has more strong effect on the disruption of IRGB10 dimer. Since K193E mutant formed more higher oligomeric peak, this phenomenon might be because of the solubility issue. Because K193E mutant is less soluble, disrupted dimer might go to higher oligomer (or aggregation) fraction. We speculated the reason of this observation in the main text.

3. Based on native PAGE electrophoresis, the authors conclude that there is no interaction between IRGB10 and GBP2 or GBP5. However, based on the literature, it is clear that the ability of these proteins to act in concert depends on the infectious context. The evidence supporting the conclusion that these proteins do not directly interact to lyse bacteria therefore appears weak. The authors should consider modifying this conclusion in the text to make the caveats more apparent or utilize further experimentation needed to conclusively make this statement. Additionally, due to smears, multiple banding patterns, and a lack of indication of molecular weights, it is difficult to interpret this data.

Response: We absolute agree with this reviewer's concern that our conclusion is too conclusive with limited experimental data. Based on this reviewer's suggestion, we modified the conclusion to

“This experiment showed that IRGB10/GBP5 mixture failed to produce new complex band on the gel, which will be produced upon protein complex formation, indicating that IRGB10 might not directly bound to GBP5 in either the absence or presence of GTP/MgCl₂ in vitro (Fig. 5c; *SI Appendix*, Fig. S2). GBP2, another protein from the GBP family, also did not produce any complex band on the gel by mixing with IRGB10 (*SI Appendix*, Fig. S3).

Although our protein interaction study indicated no direct interaction between IRGB10 and GBP family in vitro, based on the previous literature, it is clear that the ability of these proteins to act in concert depends on the infectious context {Man, 2016 #27}{Haldar, 2013 #22}. Therefore, it cannot be ruled out that the interaction between IRGB10 and GBP family might be weak, or a specific cellular condition is needed for direct interaction”

4. Please update the text to clarify that IRGA6 is also a GKS GTPase. This will help strengthen the rationale for similarities between IRGB10 and IRGA6.

Response: We added this important information at the “Comparison of the structure of IRGB10 and IRGA6 subsection” as suggested by the reviewer.

Reviewer: 3

We thank the reviewer for constructive comments on our work

Comments:

1. The introduction should contain more details on what is known from previous studies on the structure and biochemistry of IRG family members and the cellular function of IRGB10. Also a short paragraph on the structure of related dynamin-GTPases (in particular BDLP and atlastin) would be warranted to allow the reader a better assessment of the impact of this work. Furthermore, known differences in sequence and function of IRGA, IRGB and IRGM families should be worked out in detail.

Response: We agree with this reviewer's opinion that the introduction should be more informative. We added the structural and biochemical details of IRG family in the introduction. We also added a short paragraph on the structure of BDLP and atlastin as suggested by the reviewer. Finally, we compared the sequence between the same and different subfamily of IRG and provided sequence similarity information at the introduction section as suggested by the reviewer. The functions of IRGB10 with the most studied IRGM3, IRGB6, IRGA6 were already described in the introduction.

2. Crystallographic data table

R_{free} and R_{work} differ by more than 7% - the structure may be overrefined. What was the refinement strategy (what exactly was refined)? Should a more conservative refinement be considered? Please include details of the refinement strategy in the Methods. Why were phenix and REFMAC5 used in combination?

Response: We added details of the refinement strategy in the Methods, as suggested by this reviewer

3. Fig. 2: I would suggest first showing the structure of the IRGB10 monomer and then describe higher-order assemblies in the crystal. Please label the helical domains and GTPase domains in Fig. 2A.

If I understand Fig. 2E correctly, chain A from one asymmetric unit forms a physiological dimer with chain B from the adjacent asymmetric unit? If this was the case, why not place and refine the physiological dimer in the asymmetric unit, rather than using the current placement obtained randomly by MR and likely representing a non-physiological dimer.

Response: As reviewer's suggestion, we rearranged the physiologically meaningful dimer in the asymmetric unit. Based on this and reviewer #1's suggestion, we rearranged the text and Figure2 panels to reflect the fact the head-to-head dimer is the true dimer form. We also moved the figure 2B to figure 2A as suggested by this reviewer. This rearrangement is way better to follow the structural details of IRGB10. We thank this reviewer for this comment.

4. Fig. 2: The authors built a GDP molecule in the active site, although they did not add it during purification. Since the resolution of the structure is limited, they should prove by alternative methods that a GDP molecule is maintained during purification (by HPLC or mass spectrometry analysis). Can the GDP molecule be removed in the absence of magnesium, e.g. by a washing step with EDTA?

Response: Because the concentration of GDP is high in the bacterial cell, it is not surprised to have GDP in the GTPase domain. In our experience and other researches, the structures of GTPase sometimes contains GDP in the active site. Because this is 2.6 Å high resolution structure and GDP is clearly in the GTPase domain (Figure 2C) (atomic resolution structure is the best way to confirm the existence of GDP), we believe that endogenous GDP is incorporated during purification. Our preliminary data suggested that GDP was removed by addition of high salt but was not removed by EDTA washing.

5. Fig. 3D: These data are not conclusive. Based on the UV signal, different concentrations were used for wt and the two mutants. Since IRGB10 shows a concentration-dependent assembly (Fig. S1), equal concentrations must be used for this experiment. Furthermore, MALS should be employed to analyze whether the second peak in the elution profile of the mutants indeed corresponds to a monomer. Do the mutants still contain bound GDP or do they lose it during purification in the absence of the G interface? How was the protein concentration be determined? UV analysis may be misleading when nucleotides are bound differentially.

Response: It is really nice comment. We totally understand the reviewer's concern. We re-performed all the SEC-experiments by using the same amount of loaded proteins. In addition, we also performed MALS to confirm the mutagenesis effect as suggested by this reviewer. This SEC-MALS result confirmed that the second peak generated by introducing mutation is monomer. Please see the new figure 3. Based on UV analysis (A260/A280), the mutants still contained bound GDP. Protein concentration was determined by UV analysis and re-confirmed by SDS-PAGE.

6. Fig. 3. The G interface described here is important for the story and should be further analyzed. For example, does nucleotide-free IRGB10 show a GTP-dependent assembly in solution that can be measured by turbidity (similar as for IRGA6)? In this case, wt and mutants should be tested in this assay. Does IRGA6 show a concentration-dependent GTPase activity that is disturbed for the mutants?

Response: This is also important point and we wondered whether IRGB10 is further oligomerized by GTP similar with IRGA6. We already performed Native-PAGE to quick check the GTP-dependent further oligomerization of IRGB10. However, IRGB10 was not oligomerized by addition of GTP and MgCl₂ (See below).

mIRGB10 (WT) and mutant (K193E) + GTP (concentration dependent test): Native-PAGE

*All 10 mM MgCl₂ added
**K193E monomer peak was used

As suggested by reviewer, we also performed turbidity assay. This experiment also indicated that IRGB10 was not oligomerized by GTP (see below).

mIRGB10 WT & Mutants monomer : turbidity test

Based on these results, we conclude that the further oligomerization of IRGB10 is not dependent on GTP. But there might be triggering factor for IRGB10 oligomerization. The next our study will be to find this triggering factor and the function of GTP on this IRGB10.

7. Fig. 4: The sequence alignment could be moved to the supplement. Instead, some more structural comparisons of the G domain dimer in IRGA6 and IRGB10 would be useful. For example, are residues involved in dimerization conserved in IRGB10 and IRGA6? Based on this, would one predict heterodimerization? A side-by-side comparison of the IRGB10 and IRGA6 dimers with details of the interaction would be useful.

Response: We totally agree with the reviewer’s opinion. As suggested, we moved the sequence alignment to Supple info and added structural comparison of IRGB10 dimer with IRGA6 dimer.

8. Fig. 5: I found the introduced model of action interesting, but I would place it under the header 'Discussion', not 'Results'. The actual structure of a membrane-bound IRG may or may not look like the proposed model.

Response: We agree with this reviewer's opinion. We placed the model description under the header "Discussion".

9. Line 280 – What does an interface score of 0.00 and 0.25 mean?

Response: We explained the meaning of interface score produce by PISA program in the text.

10. Line 324: I think it is sufficient to say that the closest relative of IRGB10 with a known structure is IRGA6, unless the authors want to describe the (presumably very small) differences in the three published IRGA6 structures.

Response: We modified the sentence as suggested by the reviewer

11. Line 330 – root mean square deviation. Were all residues included in this analysis or only those that superimpose well ? Does the reported number correspond to Calpha superposition ?

Response: For this analysis, all the Ca atoms of IRGB10 were included. We added this information as suggested.

12. Typos:

Abstract, line 39: it decorated by guanylate binding protiens (GBPs) directly targets ...
Please correct this sentence.

Response: Corrected

Line 57 IRG are an emerging family of host defense related ...

The first manuscript on IRGs dates back to 1998 (Boehm et al., J. Immunology), e.g. 'emerging' may be misleading.

Response: Deleted "emerging"

Line 87: ... are recruited ...

Response: Corrected

Line 243: With great effort – Why with 'great effort'? Was there something particularly difficult in structure determination?

Response: We modified the sentence. Thank this reviewer for finding these typos with precise review.

Reviewers' comments:

Reviewer #1 (Remarks to the Author):

I am satisfied with the revision.

Reviewer #2 (Remarks to the Author):

In this manuscript, the authors report the crystal structure of the mouse IFN-inducible GTPase IRGB10 to a resolution of 2.6 Å and explore the functional relevance of two differentially arranged IRGB10 dimers that were observed within the crystallographic structure. While the revision has addressed several of my comments and improved some aspects of the manuscript, there are some remaining concerns.

- The authors replaced Figure 3 with spectra having a more comparable amount of protein, which was an imperative improvement. However, in the replacement Figure 3C, there is no wild-type spectra, which makes drawing conclusions difficult. Particularly because the dimer peak for the mutant appears to be near 150 kDa; this appears to be very different from what is reported for the wild-type dimer, which was measured around 80-100 kDa (text, Figure 3D, and previous figure). The authors need to add the wild-type to Figure 3C to clear up this discrepancy.
- The sentence "Based on this analysis, one tentative contact that can produce a head-to-head dimer with two-fold crystallographic symmetry was observed" from the original submission was changed to state that a tentative head-to-tail dimer was observed. I understand that the authors were restructuring this statement to reflect the soluble dimer being the head-to-head structure, but it is misleading or potentially incorrect to rephrase in this way, because one tentative contact could not have been observed for both in the crystal packing. The authors need to clarify this point.
- The grammar on the newly incorporated text needs to be improved.

Reviewer #3 (Remarks to the Author):

I am satisfied with the experimental revisions and the re-structuring of the text. On some places, the English may need a bit of polishing. I have also some more advice for improvement of the figures (no need to see them again).

Abstract:

It decorated by guanylate-binding proteins (GBPs) directly targets ... -

The sentence is still not correct – I would just remove it, it is also partly repetitive with the next sentence.

L91 – The structural studies ... - there are several mistakes in this sentence. Maybe better:

`Structural studies on dynamin superfamily members, such as Atlastin-1 and bacterial dynamin-like protein (BDLP), indicate that large-scale rearrangements between the GTPase domain and helical domain are critical for membrane binding and assembly (atlastin is not known to form a filament).

L102 – is recruited (instead of `recruit`)

L360: Since IRGA is the only IRG with a known crystal structure, and all three top hits from the DALI

search are mouse IRGA, the paragraph sounds a bit weird as it is. I would suggest something along this line:

'The top three matches from a DALI search included IRGA6 (three different structures), BDLP (xxx different structure) and Atlastin-1 (xxx different structure).' And then skip the sentence 'Among the 19-22 proteins included in the mouse family...'

L383 – head-to-head, not head-to-dead :)

Figure 1E-G: Graphically indicate in B, C, D to which fractions the MALS profiles in E, F, G belong.

Fig. 3C: For better comparison, I would include the wt profile here as well.

Fig. 3E-H: Please label in the figure which profile corresponds to which mutant.

Fig. 4: Labelling of GTPase domain and helical domain in all figures would provide a better orientation. Some headings of E, F, G to explain the content of each panel would be also helpful. A-G look quite overcrowded, a smaller secondary structure representation would make the figures clearer.

Fig. 4E-G: I am still puzzled about the comparison: The structure in Fig. 4D top appears to represent the unrelated 'head-to-tail dimer' and the structure below the related 'head-to-head' dimer (which is shown in F) – is the labelling swapped? Furthermore, the detailed differences in the two interfaces would be very informative but the corresponding detailed Figures in 4E, F bottom are just not very helpful to recognize the differences in this regard: Does the dimer interface in IRGB10 involve the nucleotide, as in IRGA6? The nucleotides are not easy to see in E, F (maybe better in a space-filling representation) and should be labelled. Where is helix H7 in F? Is the orientation the same? Please adjust this figure to make it more accessible!

Reviewers' Comments to Author:

Reviewer: 2

We thank the reviewer for his/her constructive comments on our work.

Comments:

1. The authors replaced Figure 3 with spectra having a more comparable amount of protein, which was an imperative improvement. However, in the replacement Figure 3C, there is no wild-type spectra, which makes drawing conclusions difficult. Particularly because the dimer peak for the mutant appears to be near 150 kDa; this appears to be very different from what is reported for the wild-type dimer, which was measured around 80-100 kDa (text, Figure 3D, and previous figure). The authors need to add the wild-type to Figure 3C to clear up this discrepancy.

Response: We added the wild-type profile in Figure 3C as requested by the reviewer to better comparison. As you know, the result of SEC is not quite accurate. That's why we confirmed the molecular mass changes by mutagenesis using MALS which is much more accurate method for determining molecular mass. The MALS results provided in figure 3E~F clearly indicated that D185R and K193E mutagenesis disrupted the dimer of IRGB10 and produced monomeric IRGB10.

2. The sentence "Based on this analysis, one tentative contact that can produce a head-to-head dimer with two-fold crystallographic symmetry was observed" from the original submission was changed to state that a tentative head-to-tail dimer was observed. I understand that the authors were restructuring this statement to reflect the soluble dimer being the head-to-head structure, but it is misleading or potentially incorrect to rephrase in this way, because one tentative contact could not have been observed for both in the crystal packing. The authors need to clarify this point.

Response: We modified the sentence as suggested by the reviewer.

3. The grammar on the newly incorporated text needs to be improved.

Response: We improved the newly incorporated text by editing service.

Reviewer: 3

We thank the reviewer for constructive comments on our work

Comments:

1. I am satisfied with the experimental revisions and the re-structuring of the text. On some places, the English may need a bit of polishing. I have also some more advice for improvement of the figures (no need to see them again).

Response: Thank you for your positive comments for our revised version of manuscript.

2. Abstract:

It decorated by guanylate-binding proteins (GBPs) directly targets ... -

The sentence is still not correct – I would just remove it, it is also partly repetitive with the next sentence.

Response: We removed that sentence as advised.

3. L91 – The structural studies ... - there are several mistakes in this sentence. Maybe better: 'Structural studies on dynamin superfamily members, such as Atlastin-1 and bacterial dynamin-like protein (BDLP), indicate that large-scale rearrangements between the GTPase domain and helical domain are critical for membrane binding and assembly (atlastin is not known to form a filament).

Response: We replaced it with sentence suggested by the reviewer.

4. L102 – is recruited (instead of 'recruit')

Response: We corrected it

5. L360: Since IRGA is the only IRG with a known crystal structure, and all three top hits from the DALI search are mouse IRGA, the paragraph sounds a bit weird as it is. I would suggest something along this line:

'The top three matches from a DALI search included IRGA6 (three different structures), BDLP (xxx different structure) and Atlastin-1 (xxx different structure).' And then skip the sentence 'Among the 19-22 proteins included in the mouse family...'

Response: We modified the sentence as suggested.

6. L383 – head-to-head, not head-to-dead :)

Response: We corrected this typo.

7. Figure 1E-G: Graphically indicate in B, C, D to which fractions the MALS profiles in E, F, G belong.

Response: We indicated the sample loaded for MALS

8. Fig. 3C: For better comparison, I would include the wt profile here as well.

Response: We included the wildtype profile in figure 3C as suggested.

9. Fig. 3E-H: Please label in the figure which profile corresponds to which mutant.

Response: We labeled each figure as suggested.

10. Fig. 4: Labelling of GTPase domain and helical domain in all figures would provide a better orientation. Some headings of E, F, G to explain the content of each panel would be also helpful. A-G look quite overcrowded, a smaller secondary structure representation would make the figures clearer.

Response: We labelled each domain and gave the heading for better understanding as suggested.

11. Fig. 4E-G: I am still puzzled about the comparison: The structure in Fig. 4D top appears to represent the unrelated 'head-to-tail dimer' and the structure below the related 'head-to-head' dimer (which is shown in F) – is the labelling swapped? Furthermore, the detailed differences in the two interfaces would be very informative but the corresponding detailed Figures in 4E, F bottom are just not very helpful to recognize the differences in this regard: Does the dimer interface in IRGB10 involve the nucleotide, as in IRGA6? The nucleotides are not easy to see in E, F (maybe better in a space-filling representation) and should be labelled. Where is helix H7 in F? Is the orientation the same? Please adjust this figure to make it more accessible

Response: It was mis-labelled. The newly introduced dimer interface of IRGA6 at upper panel is called to "Backside dimer". We corrected it.

We also changed the GDP to space-filling representation for better visualization. For IRGB10 dimerization, Nucleotide and H7 are involved in the dimerization of IRGB10, whereas nucleotide, P-loop, and switch I are involved in the dimerization of IRGA6. We indicated the position of H7 on the figure 4F.

REVIEWERS' COMMENTS:

Reviewer #2 (Remarks to the Author):

he authors have sufficiently addressed my remaining concerns.